# Potential Role of Photosynthesis in the Regulation of Reactive Oxygen Species and Defence Responses to *Blumeria graminis* f. sp. *tritici* in Wheat

**DOI:** 10.3390/ijms21165767

**Published:** 2020-08-11

**Authors:** Yuting Hu, Shengfu Zhong, Min Zhang, Yinping Liang, Guoshu Gong, Xiaoli Chang, Feiquan Tan, Huai Yang, Xiaoyan Qiu, Liya Luo, Peigao Luo

**Affiliations:** 1Provincial Key Laboratory of Plant Breeding and Genetics, College of Agronomy, Sichuan Agricultural University, Chengdu 611130, Sichuan, China; yutinghu@stu.sicau.edu.cn (Y.H.); zhongsicau@gmail.com (S.Z.); liangyinping3@163.com (Y.L.); guoshugong@126.com (G.G.); xl_changkit@126.com (X.C.); FeiquanTan_1@163.com (F.T.); yanghuaiv@163.com (H.Y.); 2College of Agronomy & Key Laboratory for Major Crop Diseases, Sichuan Agricultural University, Chengdu 611130, Sichuan, China; qxytzp@163.com (X.Q.); scmlsjc@163.com (L.L.)

**Keywords:** wheat powdery mildew, photosynthesis, reactive oxygen species, plant defence, salicylic acid, transcriptomics

## Abstract

Photosynthesis is not only a primary generator of reactive oxygen species (ROS) but also a component of plant defence. To determine the relationships among photosynthesis, ROS, and defence responses to powdery mildew in wheat, we compared the responses of the *Pm40*-expressing wheat line L658 and its susceptible sister line L958 at 0, 6, 12, 24, 48, and 72 h post-inoculation (hpi) with powdery mildew via analyses of transcriptomes, cytology, antioxidant activities, photosynthesis, and chlorophyll fluorescence parameters. The results showed that H_2_O_2_ accumulation in L658 was significantly greater than that in L958 at 6 and 48 hpi, and the enzymes activity and transcripts expression of peroxidase and catalase were suppressed in L658 compared with L958. In addition, the inhibition of photosynthesis in L658 paralleled the global downregulation of photosynthesis-related genes. Furthermore, the expression of the salicylic acid-related genes non-expressor of pathogenesis related genes 1 (*NPR1*), pathogenesis-related 1 *(PR1)*, and pathogenesis-related 5 (*PR5*) was upregulated, while the expression of jasmonic acid- and ethylene-related genes was inhibited in L658 compared with L958. In conclusion, the downregulation of photosynthesis-related genes likely led to a decline in photosynthesis, which may be combined with the inhibition of peroxidase (POD) and catalase (CAT) to generate two stages of H_2_O_2_ accumulation. The high level of H_2_O_2,_ salicylic acid and *PR1* and *PR5* in L658 possible initiated the hypersensitive response.

## 1. Introduction

Plants are challenged by various biotic stresses, including viral, bacterial, and fungal pathogenic stresses, which have substantial economic and ecological impacts [1,2]. Plants depend on their innate immune system to perceive and respond to biotic stimuli [3]. The first line of the immune response is called pathogen-associated molecular pattern (PAMP)-triggered immunity (PTI), which is triggered by various PAMPs [4]. During this period, the physiological and biochemical statuses of plants correspondingly change, which include phenomena such as cytoskeletal reorganization, cell wall reinforcement, reactive oxygen species (ROS) bursts, and stomatal closure [5,6,7]. To defend against pathogens further, resistance (R) proteins of plants recognize specific effectors generated by pathogens in the infection process to trigger the second line of defence referred to as effector-triggered immunity (ETI) [4]. These defence responses include expression of pathogenesis-related (*PR*) genes and initiation of the hypersensitive response (HR) to restrict biotrophic pathogen growth [4,8]. ETI may also trigger systemic acquired resistance (SAR) responses in distal uninfected tissues [9,10]. Although PTI and ETI responses are triggered by different pathogen-derived molecules, they share several downstream signals, including ROS and phytohormones [11]. The signaling overlap between PTI and ETI suggests that they are complementary, however, the differences in the strength or timing of signals lead to differences in defence strength [12]. In this process, chloroplasts, as the main production sites of ROS and phytohormones, are considered to be an important battleground for the interaction between hosts and pathogens [13,14].

In general, ROS are composed mainly of singlet oxygen (^1^O_2_), superoxide anion radical (O^2•−^), hydrogen peroxide (H_2_O_2_), and hydroxyl radical (•OH) [15,16]. In plants, ROS are produced mainly by photosystem I (PSI) and photosystem II (PSII) during photosynthesis [17]. On the one hand, ground-state oxygen (^3^O_2_) is continuously excited to produce ^1^O_2_ in the PSII reaction centre [18,19]. On the other hand, O_2_ is reduced to O^2•-^ directly by electron transport and electron leakage from the acceptor side (QA and QB) of PSI and PSII, respectively [20,21]. Finally, O^2•-^, which presents high reactivity and short life, is easily dismutated to H_2_O_2_ by superoxide dismutase (SOD) [22]. Traditionally, ROS are considered inevitably harmful by-products during aerobic metabolism. In fact, ROS, especially H_2_O_2_, not only have basic defence functions but also act as second messengers to activate the expression of defence genes [23]. Various studies have demonstrated that H_2_O_2_ could be both a signal molecule that induces defence responses at low/moderate concentrations and a defensive weapon to resist pathogen infection directly at high concentrations [16,23]. The concentration of H_2_O_2_ can be rapidly and precisely regulated by various antioxidant enzymes, mainly SOD, catalase (CAT), peroxidase (POD), and ascorbate peroxidase (APX), with CAT acting as a key H_2_O_2_-scavenging enzyme and playing a particularly crucial role in regulating H_2_O_2_ concentrations [16]. Overall, whether H_2_O_2_ acts as a signaling molecule or as a defensive weapon relies on the precise regulation between H_2_O_2_ production and scavenging, and the different interplay between H_2_O_2_-producing and H_2_O_2_-scavenging pathways during stress determines the different kinds of defence responses put forth by plants and their tolerance to stress [16,24].

In addition to H_2_O_2_, phytohormones such as salicylic acid (SA), jasmonic acid (JA), and ethylene (ET), which are mainly synthesized in chloroplasts, also play important roles in the regulation of different defence responses [13]. In host plants, the biosynthesis of ET includes the following three steps: firstly, s-adenosine Met (SAM) synthetase converts methionine (Met) into SAM; secondly, SAM is converted into 1-aminocyclopropane-1-carboxylic acid (ACC) by ACC synthase; finally, ACC oxidase (ACO) catalyses ACC to ET. The precursor for ET biosynthesis, methionine, is produced in the chloroplast [19]. In addition, linolenic acid released from chloroplast membranes is the substrate of JA biosynthesis [13]. Moreover, the isochorismate synthase (ICS) pathway, which is one of the major contributors to SA synthesis from chorismate, is localized in the chloroplast as well [25]. It is well known that there is a mutually antagonistic interaction between SA- and JA/ET-dependent signaling against distinct pathogens that have different lifestyles [4]. For example, biotrophic pathogens are inhibited mainly by SA-mediated defences, whereas defences against necrotrophic pathogens are regulated by JA/ET [26]. Furthermore, some studies have suggested that there is an interplay between plant hormones and both photosynthesis and H_2_O_2_ [24,27,28]. For instance, the expression of photosynthesis-related genes is universally downregulated under biotic stress while the expression of genes involved in the synthesis of JA, SA, and ET is upregulated, which indicates that the downregulated expression of photosynthesis-related genes is part of the defence response [29]. Furthermore, SA can inhibit H_2_O_2_-scavenging enzymes, such as CAT and APX [30,31], and in turn, H_2_O_2_ accumulation in the chloroplast can result in elevated SA levels and enhanced SA responses [32].

In addition to indirectly participating in plant defence, some photosynthesis-related proteins can directly interact with pathogens to regulate defence responses. For example, *Arabidopsis* PsbP, a PSII subunit, interacts with the coat protein of *Alfalfa* mosaic virus to inhibit viral replication [33]. The electron acceptor protein ferredoxin-I (Fd-I) is associated with the production of ROS and the HR in defence against *Pseudomonas syringae* in sweet pepper [34], and the overexpression of the *Fd-I* gene also confers enhanced resistance to virulent bacteria in other plants [35,36,37,38]. In fact, light-harvesting chlorophyll a/b-binding proteins (LHCBs) also partly modulate ROS homeostasis to affect abscisic acid (ABA) signaling [39]. Furthermore, some pathogen effectors, such as *P. syringae* HopI1 and HopN1, interact with the oxygen-evolving complex PsbQ of PSII and target chloroplast-localized heat shock protein Hsp70, respectively, to supress immunity reactions, including the suppression of SA accumulation and the reduction of ROS production and cell death [40,41]. Furthermore, some photorespiratory genes are involved in plant defence response as well [42,43]. For instance, silencing of photorespiration-related gene glycolate oxidase (*GOX*) in *Nicotiana benthamiana* increased susceptibility to *P. syringae and Xanthomonas campestris.* Furthermore, *Arabidopsis gox* mutants exhibited a reduction in H_2_O_2_ accumulation after *P. syringae* infection [44]. In tomato, *GOX2* and another two photorespiratory genes serine glyoxylate aminotransferase (*SGT*) and serine hydroxyl methyltransferase (*SHMT1*) positively regulated the defence response against *P. syringae*. H_2_O_2_ accumulation plays critical role in *GOX2*-regulated but not *SGT*-regulated and *SHMT1*-regulated SA signaling [45]. These studies indicate that photosynthesis-related proteins combined with both ROS and phytohormones play a central role in plant defence.

Wheat powdery mildew, caused by the obligate biotrophic fungus *Blumeria graminis* f. sp. *tritici* (*Bgt*), is a destructive wheat disease worldwide [46,47]. The wheat powdery mildew resistance gene *Pm40* originally derived from *Elytrigia intermedia* was mapped onto chromosome arm 7BS. This gene is associated with strong resistance against various *Bgt* strains and has been widely used in Chinese wheat breeding programmes [48,49,50]. Our pfrevious studies suggested that components associated with photosynthesis, especially the PSII reaction centre, were important sites for *Pm40*-mediated resistance, and some photosynthesis-related proteins, early ROS accumulation, papilla formation, and defence-related gene expression are involved in resistance [51,52,53].

RNA sequencing (RNA-seq) is an effective and powerful method for investigating pathogen infection, host defence and plant-pathogen interaction mechanisms. In our previous study, we compared the transcriptome of *Bgt* during interactions with the *Pm40*-expressing wheat line L658 and the *Pm40*-deficient sister line L958 to determine the pathogenesis of *Bgt* [54]. In this study, we further analysed the sequencing data with a focus on determining the resistance responses of the host from the other standpoint. In addition, differences between the two genotypes were revealed via light microscopy and antioxidant enzyme activities, and photosynthesis and chlorophyll fluorescence parameters were measured at various time points to further elucidate the relationships among photosynthesis, ROS, and plant hormones in wheat during the response to *Bgt*.

## 2. Results

### 2.1. Transcriptome Analysis of L658 and L958 during Bgt Infection

In our previous study, differences in the infection mechanism of *Bgt* during the interactions with L658 and L958 were determined [54]. To further clarify the difference in the molecular response mechanisms between L658 and L958, we continued to analyse the RNA-seq data from the plant side. Briefly, the percentage of reads mapped to the wheat reference genome was high for all samples and the smallest percentage of mapped reads reached 84.69%, with 21 out of 36 percentages higher than 90% (Appendix A). Furthermore, we screened the differentially expressed genes (DEGs) between the two genotypes at the same time point and the maximal DEG number was 2087 at 12 h post-inoculation (hpi) (Figure 1A). In addition, we also screened the DEGs in the same genotype at each inoculation time point compared with the uninoculated time point (0 h), and between two adjacent time points (Figure 1B). Interestingly, a large portion of these DEGs were the same in both L658 and L958 (shown as overlaps), and the expression trend of nearly all of them was the same (Figure 1B), which could be explained by their similar genetic background. To validate the reliability of the transcriptome data, 12 randomly selected genes were analysed via quantitative real-time PCR (qRT-PCR) (Appendix A). The correlation between normalized RNA-seq reads per kilobase per million (RPKM) values and normalized expression values from qRT-PCR was high (R = 0.8366, *p*-value = 1.12 × 10^−38^), which confirmed the reliability of RNA-seq data (Figure 1C).

### 2.2. Difference in H_2_O_2_ Accumulation and Related DEG Expression between L658 and L958

H_2_O_2_ accumulation is a typical host defence response to biotrophic fungi [4]. In this study, the results of 3′-diaminobenzidine (DAB) staining showed that H_2_O_2_ similarly accumulated under primary germ tube (PGT) at 6 and 12 hpi and then accumulated under penetration peg (PP) at 24 hpi in both L658 and L958 (Figure 2A,B). Subsequently, a stronger H_2_O_2_ response was generated in the whole infected and adjacent cells in L658 at 48 hpi, while H_2_O_2_ faded gradually in L958 at 48 and 72 hpi (Figure 2A,B). In addition, the percentage of interaction sites stained by DAB had no significant difference between L658 and L958 at 12 and 24 hpi, while that in L658 was significantly (*p* < 0.05) higher at both 6 and 48 hpi and lower at 72 hpi than that in L958 (Figure 2C). 

To further determine the difference of H_2_O_2_ at the transcript level, we identified some DEGs related to H_2_O_2_ generation and H_2_O_2_ scavenging (Table 1). In L658, the expression of H_2_O_2_ generation-related DEGs, such as oxalate oxidase (*OXO*) was upregulated, especially at 12 hpi, while the expression of many H_2_O_2_-scavenging DEGs, such as *POD* and *CAT* isozymes, was downregulated in L658 compared with L958 (Table 1). In L958, the expression of other H_2_O_2_ generation-related DEGs, including respiratory burst oxidase homologue protein (*RBOH*), polyamine oxidase (*POX*), and amine oxidase (*AOX*), were usually upregulated, while DEGs encoding POD and CAT were commonly upregulated in L958 compared with L658 (Table 1). This result indicated that although different DEGs regulated H_2_O_2_ generation in L658 and L958, the capacity to scavenge H_2_O_2_ was more seriously inhibited in L658 than L958. 

The antioxidant enzyme activities further confirmed the transcriptome data. The results showed that although the activity of SOD and POD had no significant change in L658 (Figure 2D,E), CAT activity was sharply decreased from 6 to 12 hpi (*p* < 0.01) and from 24 to 48 hpi (*p* < 0.05) (Figure 2F). In L958, the activity of SOD significantly increased from 6 to 12 hpi while the CAT activity also significantly increased from 12 to 24 hpi (Figure 2F). In contrast, the CAT activity in L958 was significantly higher (*p* < 0.01) than that in L658 at 12 and 48 hpi (Figure 2F). Therefore, the inhibition of H_2_O_2_ scavenging-related genes and enzyme activities in L658 likely caused a more abundant accumulation of H_2_O_2_ than in L958. 

### 2.3. Changes in Photosynthesis and Photosynthesis-Related Genes in Response to Bgt

Photosynthesis is not only the main generator of H_2_O_2_, but also regarded as a plant defence component [13]. To investigate the influence of *Bgt* in photosynthesis, we identified a large number of photosynthesis-related DEGs, most of which encoded chlorophyll a/b-binding proteins (Cab), proteins in reaction centres, ATP synthase (ATPase), ribulose bisphosphate carboxylase small chain/large chain (RbcS/L) and Rubisco activase (RCA) (Figure 3A). Interestingly, we found that almost all of the photosynthesis-related DEGs were downregulated in L658 compared with L958, and 54 (98%) out of 55 and 75 (96%) out of 78 photosynthesis-related DEGs were downregulated in L658 at 12 and 48 hpi, respectively (Figure 3B, Appendix A). Furthermore, 41 and 98 DEGs were also downregulated in L658 at 12 hpi and 48 hpi compared with 0 hpi, respectively, and 45 DEGs were also downregulated at 12 hpi compared with 6 hpi in L658 (Figure 3C, Appendix A). On the contrary, only eight photosynthesis-related DEGs were downregulated in L958 at both 24 and 72 hpi compared with 0 hpi, and 25 DEGs were downregulated at 24 hpi compared with 12 hpi (Figure 3C, Appendix A). This result indicated that the expression of photosynthesis-related genes in L658 was more seriously inhibited compared with that in L958, which could lead to the disruption of photosynthesis.

To further confirm the photosynthesis-related transcriptome data, we detected the photosynthetic parameters. As shown in Figure 4, the net photosynthetic rate (Pn) of L658 decreased from 0 hour (without inoculation) to 12 hpi and from 48 to 72 hpi (Figure 4A) and stomatal conductance (Gs) exhibited a reduction from 0 to 12 hpi and an increase from 12 to 72 hpi (Figure 4B). Compared with Pn, the intercellular CO_2_ concentration (Ci) of L658 increased significantly (*p* < 0.01) from 6 to 12 hpi and from 24 to 72 hpi (Figure 4C). The opposite change trend of Pn and Ci in L658 indicated that non-stomatal limitation, such as the inhibition of photosynthesis-related gene expression, was the main reason for the decline of Pn. However, the Pn, Gs, and Ci of L958 showed similar change trends, with a decrease from 0 to 12 hpi and from 48 to 72 hpi (Figure 4A–C), which indicated that the Pn of L958 decreased under stomatal limitation.

Furthermore, we used chlorophyll fluorescence to evaluate changes in PSII. In L658, the efficiency of excitation capture by open PSII reaction centres (Fv’/Fm’), actual photochemical efficiency of PSII (Φ_PSII_), electron transport rate (ETR), and coefficient of photochemical chlorophyll fluorescence quenching (qP) decreased from 0 to 12 hpi and from 24 to 72 hpi (Figure 4D–G), and the maximal photochemical efficiency of PSII in dark-adapted leaves (Fv/Fm) increased from 0 to 6 hpi and then decreased to 72 hpi (Figure 4H). Although the change tendency of Fv’/Fm’, Φ_PSII_, ETR and qP of L958 was similar to that of L658, the Φ_PSII_, ETR and qP of L958 were inhibited more strongly than those of L658 (Figure 4D–G). Interestingly, non-photochemical quenching (NPQ) significantly (*p* < 0.05) increased from 0 to 12 hpi and decreased from 12 to 72 hpi in L658, while it decreased from 0 to 6 hpi and increased from 48 to 72 hpi in L958 (Figure 4I). These data indicated that although the electron transport chain was blocked in both L658 and L958, the different changes of NPQ from 48 to 72 hpi may cause the differences in H_2_O_2_ accumulation at 48 and 72 hpi between L658 and L958.

### 2.4. Identification of DEGs Related to Phytohormones and PR Proteins

To understand the difference in hormone regulation between L658 and L958, we identified some phytohormone DEGs. The DEGs involved in SA synthesis, such as salicylic acid-binding protein (*SABP*) and isochorismate synthase (*ICS*), were downregulated in L658. However, *NPR1*, which is a downstream key regulator of SA, was upregulated in L658 at all time points (Table 2, Appendix A). In contrast, the DEGs involved in JA/ET pathways, such as the JA biosynthesis-related DEGs encoding lipoxygenase (LOX) and allene oxide synthase (AOS), and nearly all ET-related DEGs, including ethylene-responsive transcription factor (*ERF*), ethylene insensitive (*EIN*), 1-aminocyclopropane-1-carboxylate synthase (*ACS*), and 1-aminocyclopropane-1-carboxylate oxidase (*ACO*), were globally downregulated in L658, especially at 72 hpi (Table 2, Appendix A). On the contrary, the expression of *NPR1* was inhibited in L958, whereas JA/ET-related genes were highly expressed (Table 2, Appendix A). The results of the qRT-PCR analysis further confirmed that the expression of *NPR1* and *EBF1* in L658 was greater than that in L958 (Figure 5A,B), and the expression of *LOX* was lower in L658 than L958 (Figure 5C). These results showed that the SA pathway was activated and the JA/ET pathways were supressed in L658, while the JA pathway was overactivated in L958.

The dynamic expression patterns revealed that most of the DEGs encoding pathogenesis-related 1 (PR1) and pathogenesis-related 5 (PR5) were expressed at greater levels in L658, especially at 24 and 48 hpi, while nearly all DEGs encoding pathogenesis-related 10 (PR10) and pathogenesis-related 14 (PR14) were downregulated at all times in L658 (Table 3, Appendix A). However, both *PR10* and *PR14* exhibited opposite expression tendencies, with higher expression in L958 compared with L658 (Table 3, Appendix A). Furthermore, the results of the qRT-PCR analysis also confirmed that the relative expression of *PR14* was lower in L658 than in L958 at 12 and 48 hpi (Figure 5D). Overall, the high expression of *NPR1* in L658 activated the expression of *PR1* and *PR5*, which further induced host cell death (Appendix A). However, the high expression of JA/ET-related genes and *PR10* and *PR14* in L958 may have suppressed the formation of HR (Appendix A).

## 3. Discussion

Powdery mildew is a leaf disease that causes substantial injuries to host photosynthesis, which results in the reduction of photosynthesis-related protein abundance and gene expression [52,55,56]. Previously, plant defence and photosynthesis were studied separately. However, a growing number of studies suggest that photosynthesis is highly related to plant immunity to pathogens [13,14,19,42,43]. Therefore, it is meaningful to investigate the cross-talk between photosynthesis and plant immunity. Dual RNA-seq is widely used to detect the gene expression in various pathosystems, thus providing an effective way to explain the molecular mechanism of plant-fungal interactions [57]. In addition, the measurement of photosynthesis-related parameters, including the net photosynthetic rate and chlorophyll fluorescence, is a very powerful method to monitor and quantify changes in photosynthesis [58,59]. In the present study, we characterized the expression profile of photosynthesis-related genes in resistant and susceptible wheat lines by comparing the wheat transcriptome after inoculation with *Bgt*, at the same time we measured the photosynthesis-related parameters. In addition, we screened some H_2_O_2_- and defence-related genes and determined cytological reactions to reveal the relationship between photosynthesis and immunity response. 

### 3.1. Initial Stronger H_2_O_2_ Burst and Secondary Lasting H_2_O_2_ Burst Were Regulated by Antioxidant Enzyme Activities and H_2_O_2_-Related Genes and Occurred in Incompatible Reactions

ROS burst is one of the earliest immunity responses after the successful recognition of a pathogen [60]. In general, there are two distinct peaks of ROS burst induced during incompatible reactions: the first phase is often a low-amplitude and transient peak that is followed by a sustained burst of comparatively greater intensity that correlates with disease resistance; however, only the first transient burst of ROS is induced in compatible reactions [60,61,62,63]. In the present study, the first localized H_2_O_2_ burst was observed at 6 hpi, and the strength was stronger in L658 than L958 (Figure 2A–C). Subsequently, a stronger H_2_O_2_ accumulation generated around the whole infected and adjacent cells at 48 hpi in L658 while H_2_O_2_ faded gradually in L958 (Figure 2A–C). This result indicated that both the first stronger H_2_O_2_ burst and the second lasting H_2_O_2_ burst in L658 had an important role in the establishment of resistance.

To reveal the difference in H_2_O_2_ accumulation at the molecular level, we identified H_2_O_2_-related DEGs. Interestingly, the expression of the H_2_O_2_-generating DEG *OXO* was upregulated in L658 at 12 hpi (Table 1). In previous studies, OXO catalyses the degradation of oxalic acid secreted by fungi into H_2_O_2_ and is involved in H_2_O_2_ accumulation in the apoplasm during different cereal plant and fungal interactions [16,64,65]. However, powdery mildew does not secrete oxalic acid, thus, the primary role of OXO in cereals would seem to be the production of H_2_O_2_ to lignify cell walls against further fungal invasion [65,66]. Hence, DEGs encoding OXO in L658 may be involved in the first H_2_O_2_ burst to reinforce the cell wall or transmit defence signals. In addition to *OXO*, the expression of H_2_O_2_-scavenging DEGs in L658, including those encoding POD and CAT isozymes, emerged as being universally downregulated (Table 1), which corresponded with the inhibition of the POD and CAT enzyme activities in L658 (Figure 2E,F). The suppression of CAT enzyme activity promotes the increase of H_2_O_2_ concentration [29], and transgenic lines that overexpress CAT are more sensitive to pathogens because the capacity to remove H_2_O_2_ is over activated [67]. Furthermore, although the correlation between POD genes and plant defence is equivocal, PODs are generally considered to be H_2_O_2_-detoxifying enzymes that may indirectly modulate resistance by regulating H_2_O_2_ concentrations [68]. Taken together, the downregulation of *POD* and *CAT* gene expression in L658 was likely related to the suppression of enzyme activity, which further led to the increase of H_2_O_2_ accumulation. Moreover, the first H_2_O_2_ accumulation during the stage from 6 to 12 hpi was possibly involved in reinforcing the cell wall while the second lasting and stronger H_2_O_2_ accumulation from 48 to 72 hpi further induced the HR (Appendix A). 

As for L958, three DEGs encoding RBOH were upregulated (Table 1). The RBOH family is reported to mediate the production of apoplastic ROS during the defence responses, however, various RBOH proteins might have different functions in disease resistance [69,70]. For example, silencing *Rboh* of *N. benthamiana* led to a suppression of HR and increased the susceptibility to *Phytophthora infestans* [71]. However, the *Arabidopsis atrbohF* mutant exhibited more resistance to *Peronospora parasitica* and increased the HR response [72]. Therefore, whether RBOH is involved in positive or negative resistance to *Bgt* in L958 needs to be further investigated, but DEGs encoding POD and CAT were clearly upregulated in L958 compared with L658, and the enzyme activities of SOD, POD, and CAT were also significantly higher in L958 compared with L658 (Table 1, Figure 2E,F). Therefore, although other DEGs likely regulated H_2_O_2_ generation, the capacity to scavenge H_2_O_2_ was enhanced in L958 compared with L658, which caused the first weaker H_2_O_2_ burst and failure of the second H_2_O_2_ burst. 

### 3.2. Global Downregulation of Photosynthesis-Related Genes Likely Disrupted Photosynthesis and Promoted the Generation of H_2_O_2_

Generally, virulent or avirulent pathogen invasion often results in a decrease in photosynthesis to different extents, and the reduction in photosynthesis capability may represent a “hidden cost” of defence [29,73]. The plant transcriptome revealed that many photosynthesis-related genes are downregulated under biotic stress [29]. For example, photosynthesis and the expression of the photosynthesis-related transcripts, such as *Cab* and *RbcS*, were reduced in barley-*Blumeria graminis* f. sp. *hordei* (*Bgh*) incompatible interaction [74], and the downregulation of *Rubisco* and *ATPase* also occurred under other biotic stresses [10,29,75]. However, in some cases, such as in the incompatible interaction between *Arabidopsis* and *P. syringae*, the repression of photosynthesis was not correlated with the downregulation of *RbcS* and *Cab* transcripts [76], which may be related to the lack of participation of a majority of plant cells in the defence reaction, thus, the repression of *RbcS* and *Cab* cannot be detected in the whole leaves’ RNA. In the present study, the downregulation of *Cab*, *RbcS*, and *ATPase* transcripts paralleled the decline in photosynthetic rate in L658, especially from 6 to 12 hpi and from 48 to 72 hpi (Figure 3A and Figure 4A). Meanwhile, the reduction of Pn in L658 was accompanied by the decrease of Gs and the increase of Ci (Figure 4B,C), which indicated that non-stomatal limitation led to the decrease of Pn in L658 [77,78], which was consistent with the above results. Taken together, the downregulation of photosynthesis-related DEGs in L658 likely switched off photosynthesis activity to strengthen the defence reaction. Inversely, only a small part of photosynthesis-related genes in L958 were downregulated (Figure 3C). Furthermore, the Pn, Gs, and Ci decreased consistently in L958 at 12 and 72 hpi (Figure 4A–C), which suggested that the decrease of Pn in L958 was primally controlled by stomatal limitation rather than to regulate immune response. 

In addition, the expression of a DEG encoding Fd (TraesCS4A01G356200) was upregulated in L658 compared with L958 at 24 and 48 hpi (Figure 3A,B). In photosynthesis, Fd is a major element of photosynthesis-associated proteins that accept electrons from PSI and reduce NADP^+^ by ferredoxin: NADP^+^ oxidoreductase (FNR). In some plant-pathogen interactions, Fd has been suggested to participate in pathogen defence directly or to induce the production of ROS to mount the HR [29,34,38]. Therefore, it would be reasonable to assume that the upregulation of *Fd* in L658 may be involved in plant defence.

To further reveal the defence mechanism of photosynthesis, we monitored the status of PSII by detecting chlorophyll fluorescence indexes. In a previous study, *Bgh* infection led to a reduction of Φ_PSII_, an increase of NPQ, and a reduction of Fv/Fm in barley [79]. In general, the increase of NPQ is a typical host response in plant–fungus interactions to protect plants when light energy absorption exceeds the capacity for light utilization, but the decrease in NPQ predisposes chloroplasts to the production of ROS, which might be a priming mechanism for chloroplast ROS signaling during later immune responses [14,80,81]. In our study, Fv’/Fm’, Fv/Fm, Φ_PSII_, ETR, and qP significantly decreased in L658 at 12 and 72 hpi (Figure 4D–H). This response implied that photosynthetic electron transport was blocked, which could result in electron leakage in L658 for the generation of H_2_O_2_. Furthermore, the increase in NPQ in L658 at 12 hpi and the continuous decrease after 12 hpi may indicate that L658 needed photoprotection before 12 hpi but tended to generate ROS at the late stage to induce the HR (Figure 4I). In L958, the Φ_PSII_, ETR, and qP were supressed more seriously than in L658 (Figure 4E–G), which may be due to damage to the chloroplasts after *Bgt* invasion. Interestingly, the NPQ of L958 increased at 72 hpi (Figure 4I), indicating that L958 did not activate the second ROS burst and thus was insufficient to induce the HR.

### 3.3. Mutually Antagonistic Interactions between SA and JA/ET Are Involved in the Regulation of Defence Against Bgt by Regulating Different PR Genes Expression

The difference of H_2_O_2_ accumulation and photosynthesis might influence the different regulation of phytohormones, including SA, JA, and ET, after inoculation with *Bgt*. Generally, SA and JA interact antagonistically under different pathogen attacks and the SA bone fide receptor NPR1 acts as an important regulator to mediate cross-talk between SA and JA [82]. In addition, ET has positive effects on JA but may have positive or negative impacts on SA depending on the pathosystem [83,84]. Regarding the defences against biotrophic fungi, the SA signaling pathway is commonly activated and then directly regulates the conformation of NPR1 to induce the formation of SAR, the activation of *PR1* and *PR5* and to suppress the JA/ET signaling pathways [82,85,86]. In previous studies, the overexpression of *NPR1* in tobacco and apple enhanced resistance to *Botrytis cinereal* and apple powdery mildew, respectively [87,88]. In a recent study, the repression of *NPR1* in *Brassica napus* L. by RNAi significantly reduced resistance to *Sclerotinia sclerotiorum*, and silencing *NPR1* inhibited SA defence response but enhanced the JA/ET defence response [89]. In the present study, although *ICS* expression was downregulated at 12 hpi, the transcript TraesCS7A01G021800 encoding NPR1 was upregulated in L658 (Table 2, Appendix A). Compared with SA, the expression of JA biosynthesis genes, including *LOX* and *AOS*, ET biosynthesis genes, including *ACS* and *ACO*, and ET-mediated signaling genes, such as *ERF* and *EIN*, were downregulated in L658, especially at 72 hpi (Table 2, Appendix A). Furthermore, the expression of *PR1* and *PR5* was upregulated in L658 (Table 3, Appendix A). Taken together, the downregulated expression of JA/ET-related genes indicated that JA/ET levels were also decreased, which could be attributed to increased SA levels in L658. The increased SA and the upregulated expression of *NPR1*, *PR1,* and *PR5* likely induced the HR in L658.

However, in L958, JA/ET-related genes were strongly expressed (Table 2, Appendix A), which suggested that JA/ET levels increased in L958. Interestingly, the expression of the other two clusters of genes encoding PR10 and PR14 was largely upregulated in L958 as well (Table 3, Appendix A). It is reported that the expression of the *PR10* gene is induced by JA and ABA [90]. Furthermore, silencing of PR10-like proteins in *Medicago truncatula* resulted in the reduced colonization of the oomycete *Aphanomyces euteiches* and induced the expression of *PR5* [91]. In another research, a wheat orthologue of LTP, which is a member of the *PR14* family, acted as a negative regulator to resist *Puccinia striiformis* f. sp. *tritici* (Pst), and the knockdown of this gene increased wheat resistance to Pst, which was accompanied by the accumulation of H_2_O_2_ and SA [92,93]. Therefore, the increase in JA/ET level and the high expression of *PR10* and *PR14* in L958 may act as negative regulators in the defence response to promote the establishment of susceptibility.

## 4. Materials and Methods 

### 4.1. Plant and Pathogen Materials

The resistant wheat line L658 carrying *Pm40* and its susceptible sister line L958 without *Pm40* were used for comparative analyses by observations of cytological changes and determinations of physiological and biochemical indices. In addition, RNA-seq was performed for wheat plants after they were inoculated with *Bgt*, as described previously [54]. The fungus was a single-spore isolate collected from a field at Wenjiang Farm, Chengdu, Sichuan Province, and propagated weekly on the highly susceptible wheat line CY20. The wheat plants were cultivated in a growth chamber (Microclima MC1750E, Snijders Scientific, Tilburg, Holland) at 18 °C and 80% relative humidity under a 16-h light/8-h dark photoperiod. Seven-day-old seedlings of L658 and L958 were inoculated with fresh *Bgt* spores at a density of 100–200 conidia/mm^2^ by shaking infected leaf segments.

### 4.2. Cytological Observations of H_2_O_2_ and Cell Death

H_2_O_2_ accumulation was detected by the DAB (Sigma-Aldrich, Shanghai, China) staining method as described by Wang et al. [94]. The first leaves of L658 and L958 were harvested at 6, 12, 24, 48, and 72 hpi. The inoculated leaves were cut off and the end of cut leaves were immersed in 0.1% (*w*/*v*) DAB solution (pH 3.8) for 12 h to take up and react with the DAB fully. Cell death was detected by the trypan blue (Sigma-Aldrich, Shanghai, China) [53]. Leaf segments were immersed in 1% (*w*/*v*) TPB solution and then boiled for 2 min to stain the necrotic cells. Afterward, the leaf segments were decoloured in an ethanol-acetic acid (1:1, *v*/*v*) mixture until the chlorophyll was removed and then cleared in saturated chloral hydrate. The leaf segments were subsequently stored in 50% glycerol solution for examination and image collection via bright-field microscopy (Nikon Eclipse 80i, Nikon Corporation, Tokyo, Japan).

### 4.3. Determination of Antioxidant Enzyme Activity

A total of 0.2 g of fresh leaves was harvested at 6, 12, 24, 48, and 72 hpi for preparation of the crude enzyme, and samples at 0 h (without inoculation) were used as controls. Each sample was detected with three biological replicates. The fresh leaves were immediately frozen in liquid nitrogen and then homogenized in 2 mL of 50 mM pre-chilled phosphate buffer (pH 7.8) containing 1% (*w*/*v*) polyvinylpyrrolidone. The homogenate was then centrifuged at 12,000× *g* for 20 min at 4 °C, after which the supernatant was used for determination of enzyme activities. The activities of SOD, CAT, and POD were measured at 560, 240, and 470 nm, respectively, as described previously [51].

### 4.4. Determination of Photosynthesis Indices

The photosynthesis index data, including the Pn, Gs, and Ci, were collected with a portable photosynthesis system (LI-6400-02B, LI-COR, Lincoln, NE, USA) as described previously [51], and the measurement conditions included a vapour pressure deficit (VPD) of 0.6 kPa under an actinic light intensity of 1000 μmol/m^2^/s and an air temperature of 22 °C. The first measurements were performed before inoculation, and it was set as the controls (0 h). Subsequently, wheat leaves were inoculated with 100–200 conidia/mm^2^ fresh spores at 9:00 am. After 6 h (15:00 p.m.), 12 h (21:00 p.m.), 24 h (9:00 a.m.), 48 h (9:00 a.m.), and 72 h (9:00 a.m.) post inoculation, the photosynthesis indices were collected. The measurements at six time points were performed under the light period of the growth chamber. The data of six independent replications of each genotype were collected by measuring the centre of the first leaf of the wheat plants. The means of these six replications were then analysed statistically.

### 4.5. Measurements of Chlorophyll Fluorescence and Quantum Yields of PSII

Chlorophyll fluorescence was measured with a portable photosynthesis system (LI-COR 6400XT, Lincoln, NE, USA) equipped with a leaf chamber fluorometer (LI-6400-40; LI-COR, Lincoln, NE, USA) as described previously [51]. Measurements were taken from the centre of the first leaves of L658 and L958 at 0 (without inoculation), 6, 12, 24, 48, and 72 hpi. Six plants of each genotype were checked independently at the different timepoints, and the average represented the phenotypic value for subsequent statistical analysis. The data of the qP, maximum fluorescence in the light (*Fm’*), variable chlorophyll fluorescence yield in the light (*Fv’*), and Φ_PSII_ were collected in the light. Furthermore, the maximum fluorescence (*Fm*) and variable chlorophyll fluorescence yield (*Fv*) were measured with a fluorescence meter after the plants were dark adapted for 30 min.

### 4.6. RNA Extraction, cDNA Library Construction, and RNA-Seq

First leaves were randomly collected from L658 and L958 plants at 0, 6, 12, 24, 48, and 72 hpi, with each sample comprising three biological replicates. A total of 36 leaf samples were frozen in liquid nitrogen and then sent to a service company (Beijing Biomarker Technology Co, Ltd., Beijing, China) for RNA-seq. Briefly, the total RNA in wheat was extracted according to the TRIzol reagent method (Invitrogen Life Technologies, CA, USA). After the integrity and purity of the RNA was checked via an Agilent 2100 Bioanalyzer (Agilent Technologies, CA, USA) and a NanoDrop ND-1000 spectrophotometer (Thermo Scientific, DE, USA), a cDNA library was constructed by the use of a NEBNext Ultra RNA Library Prep Kit for Illumina (NEB, USA). Last, RNA-seq was performed on an Illumina HiSeq™ 2500 platform in accordance with the 150-bp paired-end sequencing strategy. All the raw sequencing data have been submitted to the NCBI Sequence Read Archive (SRA), and the accession number is SRP117269 [54].

### 4.7. Alignment of RNA-Seq Reads and Differential Gene Expression Analysis

Reads that contained an adapter or a poly-N tail or those that were of low quality were removed from the raw reads. The filtered high-quality reads were then mapped to the IWGSC RefSeq v1.0 wheat genome reference (https://wheat-urgi.versailles.inra.fr/) by HISAT2 [95], the process of which allowed up to 2 base mismatches. All identified genes were functionally annotated via BLASTX of the non-redundant (NR) database and the Swiss-Prot protein database [96]. The expression level of the transcripts was subsequently calculated and normalized to an RPKM value by StringTie [97]. Furthermore, EdgeR [98] was applied to detect DEGs in pairwise comparisons with a threshold of a false discovery rate (FDR) ≤ 0.001 and |log2(fold change) | ≥ 2.

### 4.8. Reliable Analysis by qRT-PCR

To confirm the reliability of the RNA-seq data, we randomly chose 12 expressed genes for qRT-PCR experiments. Transcript One-Step gDNA Removal and a cDNA Synthesis Supermix Kit (Transgen Biotech, Beijing, China) were used to synthesize cDNA for qRT-PCR. The primer sequences used were designed by NCBI Primer-BLAST and are listed in Additional Appendix A. The reactions were conducted on a Thermal Cycler CFX96 Real-Time System (Bio-Rad Laboratories, Hercules, CA, USA) together with SYBR Green qPCR Master Mix (Omega, Beijing, China). The relative expression of the abovementioned 12 genes was calculated according to the 2^−ΔΔ*C*t^ method [99], with *tubulin* [100] and *GAPDH* [101] used as reference genes. The experiments were performed for three biological replicates, and each biological replicate involved three technical repeats.

### 4.9. Statistical Analysis

Significant differences in the mean physiological parameters, photosynthesis parameters and relative gene expression were determined by independent sample t-tests or Duncan’s multiple range test via IBM Statistical Package for Social Science (SPSS) 19.0 software version 19.0 (SPSS Inc., Chicago, IL, USA). The data are presented as the mean values ± standard error (SEs).

## 5. Conclusions

We investigated the differences in the physiological and biochemical status and in gene expression between L658 and L958 to reveal the mechanisms of *Pm40*-regulated resistance against *Bgt*, as shown in Figure 6. During the pre-penetration stage, the expression of genes encoding OXO was upregulated in L658 to generate H_2_O_2_. At the same time, the inhibition of photosynthesis caused by the downregulated expression of genes encoding Cab, ATPase, and RbcS/L generated O^2-^, which was dismutated into H_2_O_2_ by SOD. The suppression of POD and CAT further promoted the accumulation of H_2_O_2_ in L658. The first stage of H_2_O_2_ accumulation was at 6–12 hpi, and the H_2_O_2_ generated during this stage likely acted as a signaling molecule to provoke the second and relatively more intense burst of H_2_O_2_ at 48–72 hpi. The second H_2_O_2_ accumulation possibly enhanced SA levels by directly promoting the expression of SA-related genes or by indirectly inhibiting the expression of JA-/ET-related genes. Subsequently, SA likely activated the expression of *NPR1*, *PR1,* and *PR5* and ultimately induced HR in L658.

## Figures and Tables

**Figure 1 ijms-21-05767-f001:**
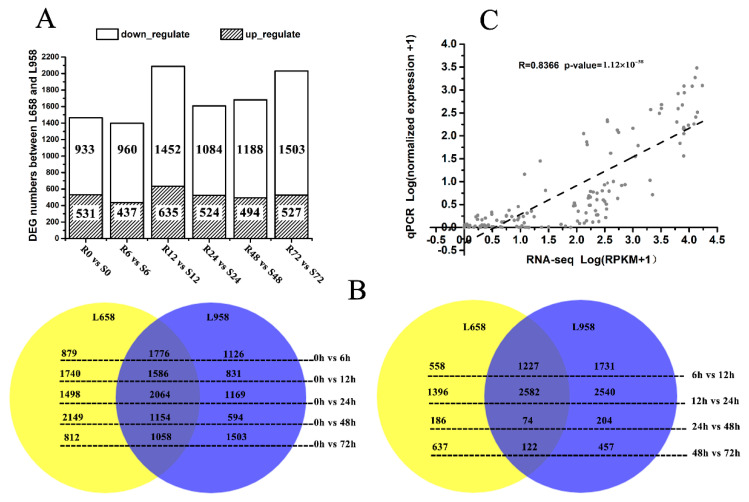
Number of differentially expressed genes (DEGs) in different comparisons and reliability of RNA-seq data as demonstrated by quantitative real-time PCR (qRT-PCR). (**A**) Number of DEGs between L658 (R) and L958 (S) at different inoculation time points; (**B**) Venn diagram showing the number of DEGs shared between (overlap) and specific to L658 (yellow) and L958 (blue) at various inoculation time points compared with 0 h (left) and between two adjacent time points (right); and (**C**) correlations between normalized RNA-seq reads per kilobase per million (RPKM) values and normalized qRT-PCR expression values. The scatterplot shows the log10 of RPKM values +1 and the log10 of qRT-PCR expression values +1; a trend line is shown as the dotted line.

**Figure 2 ijms-21-05767-f002:**
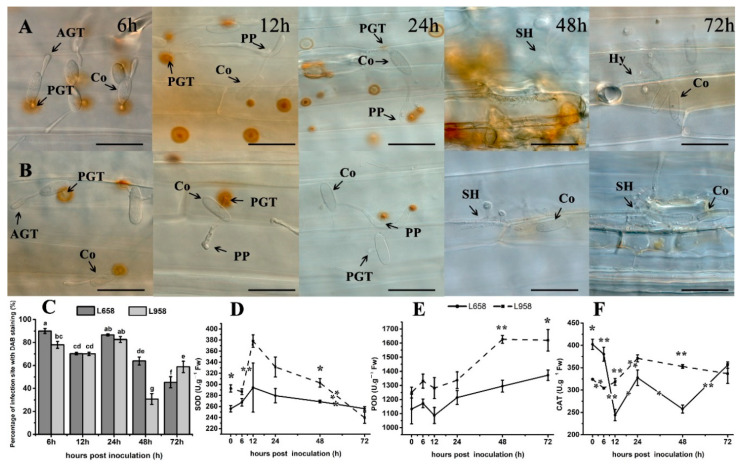
H_2_O_2_ accumulation revealed by 3’-diaminobenzidine (DAB) staining at interaction sites in the resistant wheat line L658 and susceptible wheat line L958 after inoculation with *Bgt* and activity of antioxidant enzymes at various time points. H_2_O_2_ accumulation (reddish-brown) staining by DAB at interaction sites in L658 (**A**) and L958 (**B**) at various time points. AGT: appressorium germ tube; PGT, primary germ tube; PP: penetration peg; SH: secondary hyphae; Hy, hyphae; Co: conidia. The dark bar indicates 50 μm. (**C**) Dark grey and light grey bars represent the percentage of infection sites exhibiting H_2_O_2_ accumulation in L658 and L958, respectively, after inoculation with *Bgt* at various time points. Each point represents at least 100 infection sites of each of three leaf pieces, and the lowercase letter at the top of the bar chart represents statistically significant differences at *p* < 0.05. Activities of SOD (**D**), POD (**E**), and CAT (**F**) at different inoculation time points in the two genotypes; the solid and dotted lines represent L658 and L958, respectively. The vertical bars represent the means ± SEs. The asterisks represent significant differences as follows: ** *p* < 0.01 and * *p* < 0.05. The asterisk at the top of the line chart represents the difference between L658 and L958 at each time point. The asterisk on the trend line represents the difference between two adjacent time points for the same genotype.

**Figure 3 ijms-21-05767-f003:**
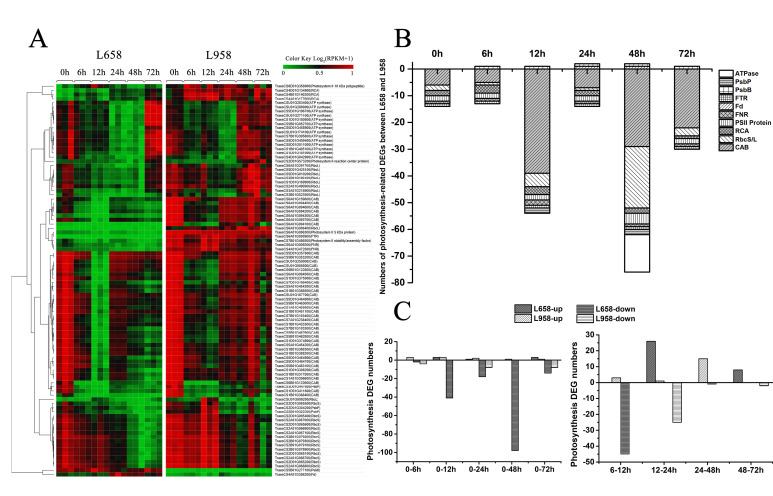
Number of DEGs and expression patterns of photosynthesis-related genes. (**A**) Different expression patterns of photosynthesis-related DEGs in L658 and L958 at various time points on the basis of log2 of RPKM values +1; the green colour represents low expression levels, and the red colour represents high expression levels. (**B**) Number of different photosynthesis-related DEGs between L658 and L958 at various time points. The negative numbers and positive numbers represent downregulated and upregulated DEGs in L658 compared with L958, respectively. ATPase: ATP synthase; FTR: ferredoxin-thioredoxin reductase; Fd: ferredoxin; FNR: ferredoxin: NADP^+^ oxidoreductase; RCA: Rubisco activase; RbcS/L: ribulose bisphosphate carboxylase small chain/large chain; Cab: chlorophyll a/b-binding protein. (**C**) Number of photosynthesis-related DEGs in L658 and L958 at various inoculation time points compared with 0 h (left) and between two adjacent time points (right).

**Figure 4 ijms-21-05767-f004:**
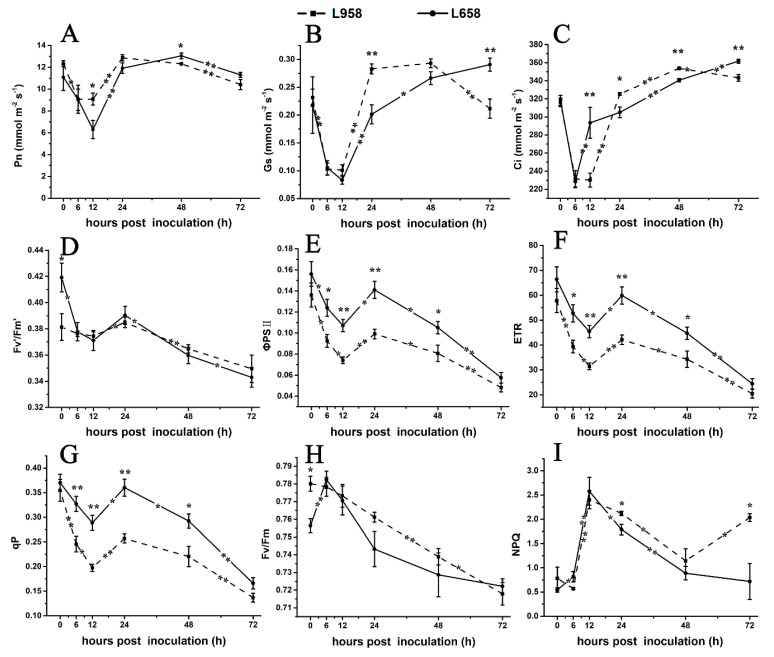
Changes in photosynthesis parameters and chlorophyll florescence parameters in L658 and L958. The solid and dotted lines represent different trends in the Pn (**A**), Gs (**B**), Ci (**C**), Fv’/Fm’ (**D**), ΦPSII (**E**), ETR (**F**), qP (**G**), Fv/Fm (**H**), and NPQ (**I**) in L658 and L958. The meanings of the symbols are the same as those in Figure 1. The asterisks represent significant differences as follows: ** *p* < 0.01 and * *p* < 0.05. The asterisk at the top of the line chart represents the difference between L658 and L958 at each time point. The asterisk on the trend line represents the difference between two adjacent time points for the same genotype.

**Figure 5 ijms-21-05767-f005:**
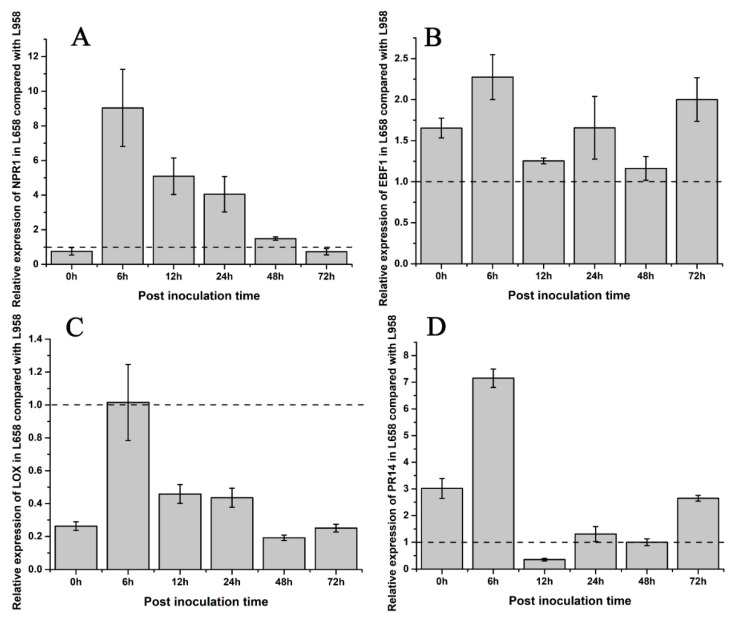
Fold change in the relative expression of defence-related genes detected by qRT-PCR in L658 compared with L958. The light grey bars represent the fold change in the relative expression of defence genes in L658 compared with L958. These genes included the following: (**A**) SA-related gene non-expressor of pathogenesis related genes 1, *NPR1*; (**B**) ethylene-related gene EIN3-binding F-box protein 1, *EBF1*; (**C**) JA-related gene lipoxygenase, *LOX*; and (**D**) pathogenesis-related protein 14, *PR14*. The dotted line means that the relative expression in L958 was set as 1.

**Figure 6 ijms-21-05767-f006:**
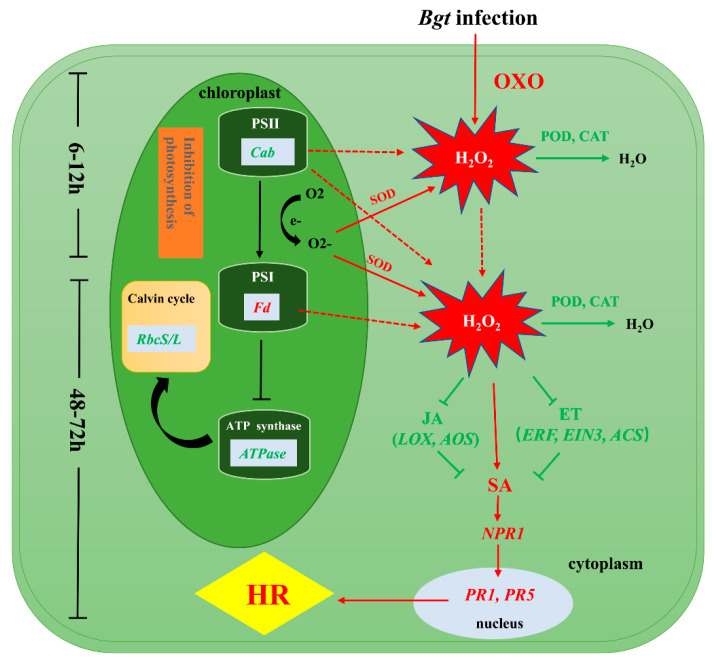
Presumptive schematic of the resistance mechanism synergistically regulated by photosynthesis, ROS and plant hormones in the *Pm40*-expressing wheat line L658. The green colour represents low expression levels, and the red colour represents high expression levels.

**Table 1 ijms-21-05767-t001:** List of DEGs involved in H_2_O_2_ production and scavenging between L658 and L958 at each time point.

Gene ID	R0-S0 ^a^	R6-S6 ^b^	R12-S12 ^c^	R24-S24 ^d^	R48-S48 ^e^	R72-S72 ^f^	Functional Annotation ^G^
**H_2_O_2_-Producing-Related DEGs**							
TraesCS6A01G180600	−6.446	−7.180	−6.906	−10.076	−7.352	−9.351	Respiratory burst oxidase homologue protein E
TraesCS5B01G299000	/	/	/	/	/	−2.425	Respiratory burst oxidase homologue protein C
TraesCS5A01G527600	/	/	/	/	/	−2.885	Respiratory burst oxidase homologue protein B
TraesCS4B01G282700	/	/	−3.409	/	/	/	Primary amine oxidase
TraesCS7D01G375700	−3.550	−3.876	−2.744	−3.351	−5.309	−6.213	Polyamine oxidase
TraesCS7A01G378800	−4.452	−4.832	/	/	−5.345	−6.612	Polyamine oxidase
TraesCS7B01G280700	2.374	/	/	/	2.694	3.339	Polyamine oxidase
TraesCS4A01G279300	/	/	/	/	/	−6.283	Oxalate oxidase GF-2.8
TraesCS4D01G032000	3.726	/	3.098	/	/	/	Oxalate oxidase GF-2.8
TraesCS4B01G033300	/	/	2.727	/	/	/	Oxalate oxidase GF-2.8
TraesCS4A01G279200	/	/	2.310	/	/	/	Oxalate oxidase GF-2.8
TraesCS4A01G279100	/	/	3.483	/	/	/	Oxalate oxidase GF-2.8
TraesCS4A01G181800	/	/	3.865	/	/	/	Oxalate oxidase GF-2.8
TraesCS4B01G033100	/	/	/	/	/	−3.112	Oxalate oxidase 2
TraesCS4D01G032200	/	/	2.298	/	/	/	Oxalate oxidase 2
TraesCS4D01G032100	−6.459	/	3.202	/	/	/	Oxalate oxidase 2
**H_2_O_2_-Scavenging-Related DEGs**							
TraesCS6B01G278100	/	/	/	−2.137	/	/	glutathione peroxidase 6,
TraesCS1B01G115800	3.734	3.949	4.754	3.230	4.803	3.701	Peroxidase N
TraesCS1A01G104300	−3.329	/	/	/	−2.280	−3.797	Peroxidase A2
TraesCS6B01G063900	/	/	/	/	/	−2.037	Peroxidase 70
TraesCS6B01G063400	/	/	/	/	/	−3.498	Peroxidase 70
TraesCS6A01G047200	/	/	/	/	/	−3.376	Peroxidase 70
TraesCS6D01G303900	/	/	−4.580	/	−4.299	/	Peroxidase 56
TraesCS6A01G324200	/	/	−3.592	/	/	/	Peroxidase 56
TraesCS1B01G115900	−4.877	−3.536	−2.526	−2.277	−2.918	−4.076	Peroxidase 54
TraesCS1D01G096400	−2.815	/	/	/	−2.264	−3.499	Peroxidase 54
TraesCS7B01G132400	/	4.140	/	/	/	2.170	Peroxidase 5
TraesCS1B01G096900	/	/	−2.658	/	/	−3.204	Peroxidase 5
TraesCS1A01G079400	/	/	−2.630	/	/	/	Peroxidase 5
TraesCS4A01G196000	/	/	−2.166	/	/	/	Peroxidase 4
TraesCS5D01G144300	/	/	/	/	/	−2.114	Peroxidase 4
TraesCS5B01G147200	/	/	/	/	/	−2.496	Peroxidase 4
TraesCS2B01G098100	/	/	/	2.053	/	/	Peroxidase 21
TraesCS3A01G297200	3.631	/	/	/	/	/	Peroxidase 2
TraesCS2B01G124600	/	−7.661	/	/	/	/	Peroxidase 2
TraesCS3A01G297100	/	/	/	/	/	−4.351	Peroxidase 2
TraesCS2D01G584600	−2.622	/	/	/	/	/	Peroxidase 12
TraesCS2D01G583200	/	/	2.692	/	/	/	Peroxidase 12
TraesCS2A01G573900	/	/	/	/	3.487	/	Peroxidase 12
TraesCS2B01G125200	/	/	7.952	/	/	/	Peroxidase 1
TraesCS2B01G125100	/	/	/	/	10.195	/	Peroxidase 1
TraesCS2D01G107900	/	−7.959	/	/	/	/	Peroxidase
TraesCS2B01G125300	/	/	/	/	/	−7.743	Peroxidase
TraesCS4A01G106300	−2.428	−2.317	/	/	/	−2.335	L-ascorbate peroxidase 1
TraesCS6A01G118300	−2.666	/	/	−2.721	−3.727	−2.990	Cationic peroxidase 1
TraesCS6A01G041700	−4.483	−4.691	−3.521	−5.265	−4.937	−5.815	Catalase isozyme 2

^a, b, c, d, e, f^: Log(fold-change) values of DEGs in L658 (R) compared with L958 (S) at 0, 6, 12, 24, 48, and 72 hpi ^G^: Putative protein function predicted based on the Swiss-Prot database.

**Table 2 ijms-21-05767-t002:** List of DEGs involved in hormone pathways between L658 and L958 at each time point.

Gene ID	R0-S0 ^a^	R6-S6 ^b^	R12-S12 ^c^	R24-S24 ^d^	R48-S48 ^e^	R72-S72 ^f^	Functional Annotation ^G^
**SA Pathway-Related DEGs**							
TraesCS3B01G354100	/	/	−3.145	/	/	/	Salicylic acid-binding protein 2
TraesCS3A01G325300	/	/	−2.796	/	/	/	Salicylic acid-binding protein 2
TraesCS3A01G325200	/	/	−3.361	/	/	/	Salicylic acid-binding protein 2
TraesCS5D01G196200	/	/	−2.201	/	/	/	Isochorismate synthase 2, chloroplastic
TraesCS7A01G021800	11.409	7.903	12.994	8.314	12.449	12.138	Regulatory protein NPR1
**JA Pathway-Related DEGs**							
TraesCS2A01G525500	/	/	/	/	/	−3.867	Seed linoleate 9S-lipoxygenase-3
TraesCS6A01G132500	−7.376	−7.356	−5.173	−7.744	−8.013	−12.116	Putative linoleate 9S-lipoxygenase 3
TraesCS6A01G132200	−13.320	−9.317	−10.259	−8.741	−14.0	−15.902	Putative linoleate 9S-lipoxygenase 3
TraesCS2D01G528500	/	/	/	/	/	−3.547	Probable linoleate 9S-lipoxygenase 5
TraesCS2B01G555400	/	−2.347	−2.003	/	−2.112	−4.107	Probable linoleate 9S-lipoxygenase 5
TraesCS6A01G181200	−9.490	−10.091	−10.025	−9.787	−9.954	−9.287	Probable linoleate 9S-lipoxygenase 4
TraesCS6B01G193400	/	/	/	−2.024	/	−2.136	Lipoxygenase 2.3, chloroplastic
TraesCS6A01G166000	−2.911	−3.863	−2.692	−2.986	−3.050	−3.241	Lipoxygenase 2.3, chloroplastic
TraesCS5D01G013400	/	−3.160	/	/	/	−4.632	Lipoxygenase 2.1, chloroplastic
TraesCS5B01G006500	/	/	/	/	/	−4.578	Lipoxygenase 2.1, chloroplastic
TraesCS5A01G007900	/	−3.314	/	/	/	−4.997	Lipoxygenase 2.1, chloroplastic
TraesCS4D01G035200	/	/	/	/	/	−2.177	Linoleate 9S-lipoxygenase 1
TraesCS4B01G037900	/	/	/	/	/	−2.780	Linoleate 9S-lipoxygenase 1
TraesCS4B01G037700	/	/	/	/	/	−2.609	Linoleate 9S-lipoxygenase 1
TraesCS4D01G238800	/	/	/	/	/	−2.213	Allene oxide synthase 2
TraesCS4D01G238700	/	−5.215	/	/	/	−6.243	Allene oxide synthase 2
TraesCS4A01G061800	/	−4.649	/	/	/	−5.560	Allene oxide synthase 2
TraesCS5D01G413200	/	/	/	/	/	−2.171	Allene oxide synthase 1, chloroplastic
**ET Pathway-Related DEGs**							
TraesCS4D01G267500	/	/	/	/	/	−2.162	Ethylene-responsive transcription factor RAP2-4
TraesCS7D01G469200	2.464	3.022	/	2.404	3.519	3.703	Ethylene-responsive transcription factor RAP2-13
TraesCS6D01G217800	/	/	/	/	/	−2.935	Ethylene-responsive transcription factor ERF053
TraesCS6B01G263800	/	/	/	/	/	−2.383	Ethylene-responsive transcription factor ERF053
TraesCS6A01G235100	/	/	/	/	/	−2.041	Ethylene-responsive transcription factor ERF053
TraesCS2A01G427700	/	/	/	/	/	−2.389	Ethylene-responsive transcription factor 7
TraesCS6A01G171900	/	/	−2.474	−2.076	/	/	Ethylene-responsive transcription factor 3
TraesCS6B01G281000	/	/	/	/	/	−4.540	Ethylene-responsive transcription factor 2
TraesCS5D01G549200	2.379	/	/	/	/	/	Ethylene-responsive transcription factor 1B
TraesCS2D01G391400	/	/	/	/	3.664	/	Ethylene-responsive transcription factor 1B
TraesCS6D01G225500	/	/	/	/	/	−4.057	Ethylene-responsive transcription factor 1
TraesCS6A01G243300	/	/	/	/	/	−2.602	Ethylene-responsive transcription factor 1
TraesCS6A01G125700	−10.243	−10.428	−11.097	−9.384	−8.635	−8.945	AP2-like ethylene-responsive transcription factor
TraesCS5A01G547500	−2.579	−9.546	−2.814	−3.525	/	−3.283	Ethylene-insensitive protein 2
TraesCS6A01G181900	−2.973	−2.685	−3.204	−3.132	−3.230	−2.830	EIN3-binding F-box protein 1
TraesCS2D01G394200	/	/	/	/	/	−3.141	1-aminocyclopropane-1-carboxylate synthase
TraesCS2B01G414800	/	/	/	/	2.471	−2.460	1-aminocyclopropane-1-carboxylate synthase
TraesCS4B01G005800	−3.100	−2.836	−3.930	−3.304	−4.105	−3.435	1-aminocyclopropane-1-carboxylate oxidase homologue 1
TraesCS4A01G499800	5.816	9.167	5.375	/	/	/	1-aminocyclopropane-1-carboxylate oxidase homologue 1
TraesCS6B01G356200	/	/	/	/	2.046	/	1-aminocyclopropane-1-carboxylate oxidase 3
TraesCS6B01G356000	/	/	/	2.029	/	/	1-aminocyclopropane-1-carboxylate oxidase 3
TraesCS5B01G232600	/	/	−2.121	/	/	/	1-aminocyclopropane-1-carboxylate oxidase 1
TraesCS5B01G232700	/	/	/	−2.167	−2.657	−2.075	1-aminocyclopropane-1-carboxylate oxidase 1

^a, b, c, d, e, f^: Log(fold-change) values of DEGs in L658 (R) compared with L958 (S) at 0, 6, 12, 24, 48, and 72 hpi. ^G^: Putative protein function predicted based on the Swiss-Prot database.

**Table 3 ijms-21-05767-t003:** List of DEGs encoding pathogenesis-related genes between L658 and L958 at each time point.

Gene ID	R0-S0 ^a^	R6-S6 ^b^	R12-S12 ^c^	R24-S24 ^d^	R48-S48 ^e^	R72-S72 ^f^	Functional Annotation ^G^
TraesCS5A01G336600	/	/	−3.422	/	/	/	Thaumatin-like protein 1a
TraesCS4D01G227400	/	/	/	/	/	−2.092	Thaumatin-like protein 1
TraesCS4A01G070700	/	/	/	/	/	−2.481	Thaumatin-like protein 1
TraesCSU01G146600	7.428	/	6.732	/	/	/	Thaumatin-like protein
TraesCS6B01G473800	8.349	5.325	10.464	/	/	/	Thaumatin-like protein
TraesCS6B01G157700	/	/	−10.474	/	/	/	Thaumatin-like protein
TraesCS6A01G129400	/	/	−9.800	/	/	/	Thaumatin-like protein
TraesCS5A01G018200	3.579	/	/	/	/	−3.910	Thaumatin-like protein
TraesCS5A01G017900	6.417	/	/	/	/	/	Thaumatin-like protein
TraesCS4A01G498000	12.101	7.376	7.329	5.273	/	/	Thaumatin-like protein
TraesCS2A01G110300	6.281	/	/	/	/	/	Thaumatin-like protein
TraesCS7D01G252400	/	4.006	/	5.578	6.710	9.868	Pathogenesis-related protein 5
TraesCS5D01G446900	7.736	/	/	/	/	/	Pathogenesis-related protein 1
TraesCS5D01G446800	8.967	/	/	/	/	/	Pathogenesis-related protein 1
TraesCS5B01G442700	7.025	/	/	/	/	/	Pathogenesis-related protein 1
TraesCS5B01G442600	8.469	/	/	/	/	/	Pathogenesis-related protein 1
TraesCS5A01G439800	9.128	/	7.573	/	/	/	Pathogenesis-related protein 1
TraesCS5A01G189200	/	/	/	3.043	3.487	/	Pathogenesis-related protein 1
TraesCS6A01G184600	−3.957	−3.644	−3.377	−4.044	−4.598	−4.097	Nuclear ribonuclease
TraesCS2D01G260000	/	/	−2.898	/	/	/	Ribonuclease 3
TraesCS6D01G320200	/	/	/	−2.215	/	−2.929	Ribonuclease 1
TraesCS6A01G339600	/	/	/	/	/	−2.331	Ribonuclease 1
TraesCS2B01G182900	/	/	/	−4.933	/	−5.114	Ribonuclease 1
TraesCS2A01G157400	/	/	/	/	/	−2.490	Ribonuclease 1
TraesCS1D01G149800	/	−2.714	−3.982	/	/	−2.658	Ribonuclease 1
TraesCS1D01G149700	/	−4.139	−5.318	/	−4.500	−5.191	Ribonuclease 1
TraesCS1B01G170200	/	−4.257	−6.333	/	/	/	Ribonuclease 1
TraesCS1B01G170100	−2.735	−5.066	−4.986	−3.003	−5.493	−6.239	Ribonuclease 1
TraesCS1A01G152800	−3.828	−4.337	−6.606	−2.809	−3.895	−4.556	Ribonuclease 1
TraesCS1A01G152600	−2.595	−5.057	−4.745	−2.907	−5.539	−6.020	Ribonuclease 1
TraesCS4B01G267300	/	/	−2.348	/	/	/	Non-specific lipid-transfer protein-like protein
TraesCS4A01G038400	/	/	/	/	/	−2.042	Non-specific lipid-transfer protein-like protein
TraesCSU01G251500	−2.325	−2.823	−4.662	−3.183	−4.485	−3.858	Non-specific lipid-transfer protein 4.3
TraesCSU01G147100	−2.496	−2.880	−4.956	−3.222	−4.536	−3.531	Non-specific lipid-transfer protein 4.3
TraesCSU01G057300	/	/	−2.685	/	/	/	Non-specific lipid-transfer protein 4.3
TraesCSU01G057200	/	/	−2.781	/	/	/	Non-specific lipid-transfer protein 4.3
TraesCS3B01G064100	−2.204	−2.562	−4.634	−2.799	−3.872	−2.701	Non-specific lipid-transfer protein 4.3
TraesCS3B01G063700	−2.223	−2.301	−4.238	−2.537	−3.851	−2.788	Non-specific lipid-transfer protein 4.3
TraesCS3B01G063100	/	/	−3.290	−2.240	−3.665	−2.777	Non-specific lipid-transfer protein 4.3
TraesCS3B01G063000	/	/	−2.874	/	−3.378	−2.383	Non-specific lipid-transfer protein 4.3
TraesCSU01G253500	−2.544	−3.492	−4.653	−3.085	−3.277	−3.503	Non-specific lipid-transfer protein 4.1
TraesCSU01G237900	−2.469	−3.284	−4.612	−3.033	−3.186	−3.472	Non-specific lipid-transfer protein 4.1
TraesCSU01G154200	−2.469	−3.291	−4.524	−2.990	−3.226	−3.472	Non-specific lipid-transfer protein 4.1
TraesCSU01G147200	−3.126	−4.274	−5.409	−3.635	−3.963	−3.991	Non-specific lipid-transfer protein 4.1
TraesCSU01G056900	/	/	/	/	/	−3.454	Non-specific lipid-transfer protein 4.1
TraesCSU01G056700	/	/	−2.123	/	/	/	Non-specific lipid-transfer protein 4.1
TraesCS3B01G064000	−2.457	−3.605	−3.873	−2.220	−2.893	−2.491	Non-specific lipid-transfer protein 4.1
TraesCS3B01G063600	/	−2.791	−3.285	/	−2.356	−2.436	Non-specific lipid-transfer protein 4.1
TraesCS3B01G063200	/	/	−4.401	−2.399	−2.962	−2.729	Non-specific lipid-transfer protein 4.1
TraesCS2A01G477700	/	/	−5.115	/	/	/	Non-specific lipid-transfer protein
TraesCS5A01G433100	/	9.147	5.006	4.132	/	4.231	Non-specific lipid transfer protein GPI-anchored 2
TraesCS3D01G331400	/	/	−2.567	/	/	/	Non-specific lipid transfer protein GPI-anchored 2

^a, b, c, d, e, f^: Log(fold-change) values of DEGs in L658 (R) compared with L958 (S) at 0, 6, 12, 24, 48, and 72 hpi. ^G^: Putative protein function predicted based on the Swiss-Prot database.

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
