# Peer review of "Potential Role of Photosynthesis in the Regulation of Reactive Oxygen Species and Defence Responses to Blumeria graminis f. sp. tritici in Wheat"

_ijms, 2020, doi:10.3390/ijms21165767_

Round 1
Reviewer 1 Report
Comments to MS “Potential role of photosynthesis in the regulation of reactive oxygen species and defence responses to Blumeria graminis f. sp. tritici in wheat” written by Yuting Hu et al.
This manuscript focuses on changes in some biochemical/biophysical parameters in two wheat lines: susceptible and resistant to pathogen. The analysis of the effects caused by biotrophic and necrotrophic pathogens on plants with focus on photosynthetic electron transport and genes related to photosynthetic machinery and ROS production was described earlier. It was demonstrated many times that photochemic/photosynthetic activity play an important regulatory role in acclimation and defense mechanisms in all plants. We know some details about regulation of ETR, especially PSII activities and ROS level, however, our knowledge about the role of chosen genes in some important phases after inoculation is not complete.
The problem is well stated and the general intention is good. There are several novel aspects of this work that extend beyond what has previously been published in the literature. It is really interesting subject in plant physiology. However, there are some issues author should address before this manuscript could be considered acceptable.
Major criticism
Introduction:
P2 L 55-56 It is not correct to write that both species of reactive oxygen species are dismutated by SOD.
P2 L 69-70 Our knowledge about localization of ET synthesis within the cell is more detailed. This should be added.
P2 L 82-85 Our knowledge on the expression of photosynthesis-related genes is broader than described here. Some facts related to photorespiratory processes could be described also.
CO2 fixation with help of PEP-carboxylase taking place also in C3 plants could be related to results shown in this manuscript. I would suspect that among 2087 genes (as indicated in this manuscript) analyzed in this manuscript genes responsible for β-carboxylation were also changed. It is necessary to add several sentences describing the potential role of these proteins in analyzed plants. Also recently presented papers on H2O2 amount in plant tissues, with CO2 concentrating mechanism (Mesembryanthemum crystallinum) in C3 and CAM state, could support presented idea. Published papers with necrotrophic (Botrytis cinerea) and biotrophic (Pseudomonas syringae) could add some interesting ideas helping to understand obtained results. To my opinion there is also some indication that presented in this manuscript mechanism could work in other plants.
M&M
P.15 L453
Was determination of photosynthesis indices made on growing plants? Were they infected?
This should be indicated why other analyses were made on leaf segments.
Generally, the paper is concisely written. After correction this can be very nice contribution to MDPI-Molecular sciences.
Author Response
Response to Reviewer 1
Major criticism
Introduction:
P2 L 55-56 It is not correct to write that both species of reactive oxygen species are dismutated by SOD.
Response: Thank you. We have deleted 1O2 and modified this sentence in line 55.
P2 L 69-70 Our knowledge about localization of ET synthesis within the cell is more detailed. This should be added.
Response: Thank you, and we have added the description about ET synthesis in line 71-74.
P2 L 82-85 Our knowledge on the expression of photosynthesis-related genes is broader than described here. Some facts related to photorespiratory processes could be described also.
CO2 fixation with help of PEP-carboxylase taking place also in C3 plants could be related to results shown in this manuscript. I would suspect that among 2087 genes (as indicated in this manuscript) analyzed in this manuscript genes responsible for β-carboxylation were also changed. It is necessary to add several sentences describing the potential role of these proteins in analyzed plants. Also recently presented papers on H2O2 amount in plant tissues, with CO2 concentrating mechanism (Mesembryanthemum crystallinum) in C3 and CAM state, could support presented idea. Published papers with necrotrophic (Botrytis cinerea) and biotrophic (Pseudomonas syringae) could add some interesting ideas helping to understand obtained results. To my opinion there is also some indication that presented in this manuscript mechanism could work in other plants.
Response: Thank you very much for your advice. The function of photorespiratory in plant defense has been introduced in revision manuscript line 110-118. It is reasonable to speculate that the genes encoding PEP-carboxylase changed in our study. However, we found only one gene encoding PEP-carboxylase (TraesCS6A01G195600) downregulated in L658 compared with L958 at various time points (including 0 hour). So, we are not sure if this change is caused by pathogen infection or the difference between L658 and L958 itself. But we still believe this is a very valuable research direction to have a better understanding of wheat-powdery mildew interaction mechanism, and we will study on it in future.
M&M
P.15 L453
Was determination of photosynthesis indices made on growing plants? Were they infected?
This should be indicated why other analyses were made on leaf segments.
Response: The measurement of photosynthesis indices was performed on growing plants before inoculation (0h) and at 6, 12, 24, 48, 72 hours post inoculation. We have added the statements in 4.4 section of Materials and Methods.
Reviewer 2 Report
In the article “Potential Role of Photosynthesis in the Regulation of Reactive Oxygen Species and Defence Responses to Blumeria graminis f. sp. tritici in Wheat” authors Yuting Hu, Shengfu Zhong, Min Zhang, Yinping Liang, Guoshu Gong, Xiaoli Chang, Feiquan Tan, Huai Yang, Xiaoyan Qiu, Liya Luo and Peigao Luo use different approaches/methods for researching interaction between the pathogen and plant physiology, mainly connected to photosynthesis. The interplay between pathogens (or stress in general), plant hormones, reactive oxygen species, gene transcription and photosynthesis has been studied for a very long time by many authors and is still largely undefined, leaving a very challenging field for further scientific research. The authors of this article have given quite clear observation of events regarding H2O2, some plant hormone metabolism and signalling, and change in gene transcript abundance as a response to Blumeria graminis in wheat resistant to the wheat powdery mildew and non-resistant one. Discussion is very good written, answering to the all of the open questions remaining after reading result section. In my opinion, the manuscript is worth publishing in IJMS, just with some minor changes.
Line 51. hydroxyl radicals (•OH), not (•OHs)
Line 69. jasmonic acid (JA), and ethylene
Line 79. “there is a interplay” instead of “there are interplays”
Figure 1A is low quality. X-axis legend is not visible. The figure quality has to be improved for publication
Line 141. dotted line is not blue, as stated in the figure legend
Line 190. Bad sentence “As the main generator of H2O2, photosynthesis is also regarded as a plant defence component.” What is the meaning of it? Please reformulate.
Line 206. Was Pn measured always at the same time (in the day cycle). When did the infection occur? O h time point should be defined here and in materials and methods – was it during the light period or during the dark? Plant physiology is not the same throughout the day. It matters a lot when the photosynthetic parameters are measured. Please indicate time points for all measurements.
Line 228. Figure 3B – it is very hard to distinguish patterns on the graphs. E.g. FNR and RCA at 72 hpi, or PsbB and PSII protein at 48 hpi
Figure 3C right is not clear – why is there no both L658 and L958 at each time point?
Line 265. PR proteins are not explained. PR1 and PR5 are explained in the Abstract section, but PR10 and PR14 are not explained at all. All PR proteins should be explained (by name and meaning) here in text. Later, in Fig. 5, it is clearly stated what each of them mean, but it is too far away in the manuscript.
Line 286. Cross-talk between photosynthesis and plant immunity is a very complex research topic. In my opinion, the authors should cite much more studies in this section.
Line 325. The authors repeat the statement as in line 314/315
Line 340. “(TraesCS4A01G356200)” should go after “encoding Fd”
Line 366. The proper name for FNR would be ferredoxin:NADP+ oxydoreductase. Please change throughout the manuscript.
Line 373. What excess light energy are the authors talking about? Please clarify.
Line 382. The statement is too speculative. Has it been shown that Bgt invasion causes destruction of host thylakoids?
Line 432. Please explain hpi the first time it is used in text (hours post infection)
“Three inoculated leaves were cut at 8 h before sampling, after which…” Could you please explain what “8 h before sampling” means? It is unclear.
Line 529. “MZ,” “LL, and HY”
Line 530. “; YH and PL”
Line 340. Abbreviations – please add PR1, PR5, PR10, PR14
- FNR should be ferredoxin:NADP+ oxydoreductase
References:
Line 543. 2. Which Journal?
Line 569. 16. Volume? Pages?
Line 576. 19. Remove extra space
Line 590. 25. Remove extra space
Line 601. 30. Remove extra space
Line 622. 37. Remove extra space
Line 636. Name of publication, web page?
Line 703. PNAS is shortage of the Journal
Line 751. 88. If it is in Journal please add the name. If it is a book please add Editors and Publisher
Author Response
Response to Reviewer 2
Line 51. hydroxyl radicals (•OH), not (•OHs)
Response: Thank you, and we have changed it in line 51.
Line 69. jasmonic acid (JA), and ethylene
Response: We have added comma in line 69.
Line 79. “there is a interplay” instead of “there are interplays”
Response: Thank you, and we have changed it in line 81.
Figure 1A is low quality. X-axis legend is not visible. The figure quality has to be improved for publication
Response: We have modified the form of X-axis of Figure 1A.
Line 141. dotted line is not blue, as stated in the figure legend
Response: We have changed it to “dotted line” in line 169.
Line 190. Bad sentence “As the main generator of H2O2, photosynthesis is also regarded as a plant defence component.” What is the meaning of it? Please reformulate.
Response: Thank you, and we have rewritten this sentence in line 220.
Line 206. Was Pn measured always at the same time (in the day cycle). When did the infection occur? 0 h time point should be defined here and in materials and methods – was it during the light period or during the dark? Plant physiology is not the same throughout the day. It matters a lot when the photosynthetic parameters are measured. Please indicate time points for all measurements.
Response: Thank you for your suggestions. In section 4.4 of Material and Method of the revision manuscript, we have declared all points that you mentioned above.
Line 228. Figure 3B – it is very hard to distinguish patterns on the graphs. E.g. FNR and RCA at 72 hpi, or PsbB and PSII protein at 48 hpi
Figure 3C right is not clear – why is there no both L658 and L958 at each time point?
Response: They are not easy to distinguish due to the numbers of these genes were very low. To have a better understanding, we have added Table S3 and Table S4 to show the specific numbers of these up/down-regulated genes in L658 and L958. Table S4 can explain why there is no genes in both L658 and L958 at some time points in Figure 3. Such as the “6 vs 12” group, there was no downregulated DEG in L958 and no upregulated DEG in L658 at 12 hpi compared with 6 hpi.
Line 265. PR proteins are not explained. PR1 and PR5 are explained in the Abstract section, but PR10 and PR14 are not explained at all. All PR proteins should be explained (by name and meaning) here in text. Later, in Fig. 5, it is clearly stated what each of them mean, but it is too far away in the manuscript.
Response: Thank you, and we have added the explanation of PR10 and PR14 in line 296 where they first appeared.
Line 286. Cross-talk between photosynthesis and plant immunity is a very complex research topic. In my opinion, the authors should cite much more studies in this section.
Response: Thank you for your suggestion. We have cited more papers in line 322.
Line 325. The authors repeat the statement as in line 314/315
Response: Thank you, the sentence “Hence, DEGs encoding OXO in L658…transmit defece signals” summarized the putative function of OXO (likely to generate H2O2 to reinforce cell wall or transmit defense signal) during 6hpi to 12 hpi. While, the sentence “Moreover, the first H2O2 accumulation…further induced the HR” subscribed the dual role of H2O2 during different stages. Therefore, the first sentence explained the cause of H2O2 generation and the second sentence described the result caused by H2O2.
Line 340. “(TraesCS4A01G356200)” should go after “encoding Fd”
Response: Thank you, we have changed the sequence in line 421.
Line 366. The proper name for FNR would be ferredoxin:NADP+ oxydoreductase. Please change throughout the manuscript.
Response: Thank you, and we have changed the name of FNR throughout the revision manuscript.
Line 373. What excess light energy are the authors talking about? Please clarify.
Response: The excess light energy means that the extra light energy when light energy absorption exceeds the capacity for light utilization. We have clarified the definition in line 431-432.
Line 382. The statement is too speculative. Has it been shown that Bgt invasion causes destruction of host thylakoids?
Response: Thank you, and we have deleted the unclear part in line 440.
Line 432. Please explain hpi the first time it is used in text (hours post infection)
“Three inoculated leaves were cut at 8 h before sampling, after which…” Could you please explain what “8 h before sampling” means? It is unclear.
Response: Thank you, we have explained “hpi” in line 146 when the first time it is used. The dyeing method has been redefined in revision manuscript line 508.
Line 529. “MZ,” “LL, and HY”
Response: We have changed it in line 634.
Line 530. “; YH and PL”
Response: We have changed it in line 635.
Line 340. Abbreviations – please add PR1, PR5, PR10, PR14
FNR should be ferredoxin:NADP+ oxydoreductase
Response: We have added them.
References:
Line 543. 2. Which Journal?
Line 569. 16. Volume? Pages?
Line 576. 19. Remove extra space
Line 590. 25. Remove extra space
Line 601. 30. Remove extra space
Line 622. 37. Remove extra space
Line 636. Name of publication, web page?
Line 703. PNAS is shortage of the Journal
Line 751. 88. If it is in Journal please add the name. If it is a book please add Editors and Publisher
Response: Thank you for checking the references. We have modified the format according to the request of IJMS.